# Decoding Rewards in Competitive Games:
# Inverse Game Theory with Entropy Regularization

**Junyi Liao** [1] **Zihan Zhu** [2] **Ethan X. Fang** [1] **Zhuoran Yang** [3] **Vahid Tarokh** [1]

## Abstract

Estimating the unknown reward functions driving agents' behavior is a central challenge in inverse games and reinforcement learning. This paper introduces a unified framework for reward function recovery in two-player zero-sum matrix games and Markov games with entropy regularization. Given observed player strategies and actions, we aim to reconstruct the underlying reward functions. This task is challenging due to the inherent ambiguity of inverse problems, the non-uniqueness of feasible rewards, and limited observational data coverage. To address these challenges, we establish reward function identifiability using the quantal response equilibrium (QRE) under linear assumptions. Building on this theoretical foundation, we propose an algorithm to learn reward from observed actions, designed to capture all plausible reward parameters by constructing confidence sets. Our algorithm works in both static and dynamic settings and is adaptable to incorporate other methods, such as Maximum Likelihood Estimation (MLE). We provide strong theoretical guarantees for the reliability and sample-efficiency of our algorithm. Empirical results demonstrate the framework's effectiveness in accurately recovering reward functions across various scenarios, offering new insights into decision-making in competitive environments.

## 1. Introduction

Understanding the underlying reward functions that drive agents' behavior is a central problem in inverse reinforcement learning (IRL) (Ng & Russell, 2000; Arora & Doshi, 2020). While traditional reinforcement learning (RL) (Szepesvári, 2010; Sutton & Barto, 2018) focuses on solving policies based on a known reward function, IRL inverts this process, aiming to infer the reward function from observed behavior. In competitive settings, such as two-player zero-sum games, this problem becomes even more complicated, as the agents' strategies depend not only on their own rewards but also on their opponents' strategies (Wang & Klabjan, 2018; Savas et al., 2019; Wei et al., 2021). These challenges motivate the study of inverse game theory (Lin et al., 2014; Yu et al., 2019), which seeks to recover reward functions from observed strategies in competitive games.

From a practical perspective, inferring the reward functions in competitive games has wide-ranging applications in economics, cyber security, robotics, and autonomous systems (Ng & Russell, 2000; Ziebart et al., 2008). Understanding the motivations behind players' actions in adversarial settings help optimize resource allocation in cyber security (Miehling et al., 2018), model strategic interactions in economic markets (Chow & Djavadian, 2015), or design better AI systems for competitive tasks (Huang et al., 2019).

Meanwhile, recovering reward functions in competitive games involves several key challenges: (i) Inverse problems are inherently ill-posed (Ahuja & Orlin, 2001; Yu et al., 2019), as multiple reward functions can lead to the same optimal strategy and equilibrium solutions. A well-designed algorithm should not merely recover a single reward function but instead identify the entire set of feasible reward functions (Metelli et al., 2021; Lindner et al., 2022; Metelli et al., 2023). (ii) In an offline setting (Jarboui & Perchet, 2021), insufficient dataset coverage is also a significant challenge. Observed strategies often fail to comprehensively cover the state-action space, making it difficult to ensure robust reward function recovery. These challenges are further amplified in Markov games (Littman, 1994), where agents' strategies evolve dynamically over time, introducing additional complexity in reward identification and estimation.

### 1.1. Major Contributions

We propose a unified framework for inverse game theory that addresses the identification and estimation of reward

---

[1]Department of Electrical and Computer Engineering, Duke University, Durham NC, USA [2]Department of Statistics and Data Science, University of Pennsylvania, Philadelphia PA, USA [3]Department of Statistics and Data Science, Yale University, New Haven CT, USA. Correspondence to: Junyi Liao <junyi.liao@duke.edu>.

*Proceedings of the 42$^{nd}$ International Conference on Machine Learning*, Vancouver, Canada. PMLR 267, 2025. Copyright 2025 by the author(s).

functions in competitive games in both static and dynamic settings. Our contribution is four-fold:

- Identification of Reward Functions: We study the identification problem using the quantal response equilibrium (QRE) under a linear assumption. We formally define the conditions for reward parameter identifiability and characterize the feasible set when parameters are not uniquely identifiable.

- Algorithm for Reward Estimation: Building on the identification results, we propose an algorithm that estimates reward functions by constructing confidence sets to capture all feasible reward parameters.

- Extension to Markov Games: We extend our framework to entropy-regularized Markov games, combining reward recovery with transition kernel estimation to handle dynamic settings. This approach is designed to be sample-efficient and adaptable, incorporating methods like Maximum Likelihood Estimation (MLE).

- Theoretical and Empirical Validation: We provide rigorous theoretical guarantees to establish the reliability and efficiency of our algorithm. Additionally, numerical experiments demonstrate the effectiveness of our framework in accurately recovering reward functions across various competitive scenarios.

## 1.2. Related Work

**Zero-sum Markov Games.** The zero-sum Markov game (Shapley, 1953; Xie et al., 2020; Cen et al., 2023; Kalogiannis & Panageas, 2023) models the competitive interactions between two players in dynamic environments. The solution typically focuses on finding equilibrium strategies (Nash Jr, 1951; McKelvey & Palfrey, 1995; Xie et al., 2020) where neither player can unilaterally improve their outcome. With a primary focus on learning in a sample-efficient manner, learning algorithms are proposed, including policy-based methods (Cen et al., 2021; Wei et al., 2021; Zhao et al., 2022; Cen et al., 2023) and value-based methods (Xie et al., 2020; Chen et al., 2022; Kalogiannis & Panageas, 2023).

**Inverse Optimization and Inverse Reinforcement Learning (IRL).** Inverse optimization (Ahuja & Orlin, 2001; Chan et al., 2022; Ahmadi et al., 2023) reverses the traditional optimization process by taking observed decisions as input to infer an objective function (Ahuja & Orlin, 2001; Nourollahi & Ghate, 2018) and constraints (Chan & Kaw, 2019; Ghobadi & Mahmoudzadeh, 2021) that make these decisions approximately or exactly optimal. In practice, inverse optimization offers a powerful framework for understanding and modeling decision-making in complex systems across fields like marketing (Chow & Djavadian, 2015; Vatandoust et al., 2023), operations research (Brotcorne et al., 2005; Agarwal & Özlem Ergun, 2010; Yu et al., 2021),

and machine learning (Konstantakopoulos et al., 2017; Dong et al., 2018; Tan et al., 2019).

Inverse reinforcement learning (Ng & Russell, 2000; Ziebart et al., 2008; Herman et al., 2016; Wulfmeier et al., 2016; Arora & Doshi, 2020) focuses on inferring the reward function based on the observed behavior or strategy of agents and experts, which is crucial for understanding various decision-making processes, from single-agent processes (Boularias et al., 2011; Herman et al., 2016; Fu et al., 2018) to competitive or cooperative games (Vorobeychik et al., 2007; Ling et al., 2018; Wang & Klabjan, 2018; Wu et al., 2024). A popular approach within the field of IRL is the Maximum Entropy IRL (Ziebart et al., 2008; Ziebart, 2018; Wulfmeier et al., 2016; Snoswell et al., 2020), which is based on the principle of maximum entropy and is provably efficient in handling uncertainty of agent behaviors (Snoswell et al., 2020; Gleave & Toyer, 2022) and high-dimensional observations (Wulfmeier et al., 2016; Snoswell et al., 2020; Song et al., 2022).

**Entropy Regularization in RL and Games.** We use the entropy regularization in our framework, which has become a widely used technique in reinforcement learning (Szepesvári, 2010; Ziebart, 2018) and game theory (Savas et al., 2019; Guan et al., 2021; Cen et al., 2023). Entropy regularization is provably effective in addressing challenges like exploration-exploitation tradeoff (Haarnoja et al., 2018; Wang et al., 2019; Ahmed et al., 2019; Neu et al., 2017), algorithm robustness (Zhao et al., 2020; Guo et al., 2021) and convergence acceleration (Cen et al., 2021; Cen et al., 2023; Zhan et al., 2023). Importantly, entropy regularization has also been shown to improve identifiability in inverse reinforcement learning (IRL) problems. Recent works in single-agent IRL, such as Cao et al. (2021) and Rolland et al. (2022), leverage entropy-regularized policies to transform ill-posed IRL problems into identifiable ones under mild assumptions. Our work builds on this insight by extending it to competitive multi-agent settings, where identifiability becomes even more subtle due to strategic interactions.

**Paper Organization.** In §2, we develop the framework of inverse game theory for entropy-regularized zero-sum games. In §3, we extend the framework introduced in §2 to a sequential decision-making setting, focusing on entropy-regularized zero-sum Markov games. We provide numerical experiments to validate the theoretical findings in §4, and conclude the paper in §5.

**Notations.** We introduce some useful notation before proceeding. Throughout this paper, we denote the set $1, 2, \cdots, n$ by $[n]$ for any positive integer $n$. For two positive sequences $(a_n)_{n=1}^{\infty}$ and $(b_n)_{n=1}^{\infty}$, we write $a_n = \mathcal{O}(b_n)$ or $a_n \lesssim b_n$ if there exists a positive constant $C$ such that $a_n \leq C \cdot b_n$. For any integer $d$, we denote the $d$-

dimensional Euclidean space by $\mathbb{R}^d$, with inner product $\langle x, y \rangle = x^\top y$ and the induced norm $\|x\| = \sqrt{\langle x, x \rangle}$. For any matrix $A = (a_{ij})$, the Frobenius norm of $A$ is $\|A\|_{\mathrm{F}} = (\sum_{i,j} a_{ij}^2)^{1/2}$, and the operator norm (or spectral norm) of $A$ is $\|A\|_{\mathrm{op}} = \sigma_1(A)$, where $\sigma_1(A)$ stands for the largest singular value of $A$. For any square matrix $A = (a_{ij})$, denote its trace by $\mathrm{tr}(A) = \sum_i a_{ii}$. For a nonempty set $\mathcal{X}$, we denote by $\Delta(\mathcal{X})$ the space of all probability distributions on $\mathcal{X}$.

## 2. Entropy-Regularized Zero-Sum Matrix Games

We derive the inverse game theory for entropy-regularized two-player zero-sum matrix games. We consider the identification problem of payoff matrices under the linear parametric assumption and derive a necessary and sufficient condition for strong identification. Furthermore, we propose methods to recover identified sets and payoff matrices.

### 2.1. Preliminary and Problem Formulation

We consider a two-player zero-sum matrix game, which is specified by a triple $(\mathcal{A}, \mathcal{B}, Q)$, where $\mathcal{A} = \{1, 2, \cdots, m\}$ and $\mathcal{B} = \{1, 2, \cdots, n\}$ are finite sets of actions that players $i \in \{1, 2\}$ can take, and $Q(\cdot, \cdot)$ is the payoff function. The zero-sum game can be formulated as the following min-max optimization problem

$$\max_{\mu} \min_{\nu} \mu^\top Q \nu,$$

where $\mu \in \Delta(\mathcal{A})$ and $\nu \in \Delta(\mathcal{B})$ are policies for each player, and $Q = (Q(a, b))_{a \in \mathcal{A}, b \in \mathcal{B}} \in \mathbb{R}^{m \times n}$ denotes the payoff matrix. The solution of this optimization problem is also known as the Nash equilibrium (Nash Jr, 1951), where both agents play the best response against the other agent.

**Entropy-Regularized Two-Player Zero-Sum Matrix Game.** We study the entropy-regularized matrix game. Formally, this amounts to solving the following matrix game with entropy regularization (Mertikopoulos & Sandholm, 2016):

$$\max_{\mu} \min_{\nu} \mu^\top Q \nu + \eta^{-1} \mathcal{H}(\mu) - \eta^{-1} \mathcal{H}(\nu),$$

where $\eta > 0$ is the regluarization parameter, and

$$\mathcal{H}(\pi) = -\sum_i \pi_i \log(\pi_i)$$

denotes the Shannon entropy (Shannon, 1948) of $\pi$. According to the von-Neumann minimax theorem (von Neumann, 1928), there exists a unique solution $(\mu^*, \nu^*)$ to this min-max problem, denoted as the quantal response equilibrium

(McKelvey & Palfrey, 1995), which satisfies the following fixed point equations:

$$\begin{cases} \mu^*(a) = \dfrac{e^{\eta Q(a, \cdot) \nu^*}}{\sum_{a \in \mathcal{A}} e^{\eta Q(a, \cdot) \nu^*}}, & \text{for all } a \in \mathcal{A}, \\[2mm] \nu^*(b) = \dfrac{e^{-\eta Q(\cdot, b)^\top \mu^*}}{\sum_{b \in \mathcal{B}} e^{-\eta Q(\cdot, b)^\top \mu^*}}, & \text{for all } b \in \mathcal{B}. \end{cases}$$

This non-linear system is equivalent to the following $m + n - 2$ linear constraints: for all $a \in \mathcal{A}$ and $b \in \mathcal{B}$,

$$\begin{cases} (Q(a, \cdot) - Q(1, \cdot)) \nu^* = \log(\mu^*(a)/\mu^*(1))/\eta, \\ (Q(\cdot, b) - Q(\cdot, 1))^\top \mu^* = -\log(\nu^*(b)/\nu^*(1))/\eta. \end{cases} \quad (1)$$

**Goal.** We study the inverse game theory for this entropy-regularized zero-sum game. To elaborate, we observe strategy pairs $(a^k, b^k) \overset{\mathrm{iid}}{\sim} (\mu^*, \nu^*)$ follows the QRE, and we aim to recover all the feasible payoff functions $Q(\cdot, \cdot)$.

**Identification of payoff matrices.** To derive inverse game theory, it is important to study the identifiability of the payoff matrix, i.e. if there exists a unique payoff matrix that satisfies the QRE constraint. In this paper, we study the identification problem under the linear structure assumption (§2.2) and further generalize the analysis to the partial identification case (§2.3).

### 2.2. Strong Identification

Suppose $(\mu^*, \nu^*)$ are the QRE for two players and we use the observed data to obtain an estimation denoted by $(\widehat{\mu}, \widehat{\nu})$. Next, we are going to estimate the payoff matrix from this estimated QRE. To ensure the game is identifiable, we leverage the following linear parametric assumption.

**Assumption 2.1** (Linear payoff functions). Suppose that there exists a vector-valued kernel $\phi : \mathcal{A} \times \mathcal{B} \to \mathbb{R}^d$ and a vector $\theta^* \in \mathbb{R}^d$ such that $\|\theta^*\| \leq M$ for some $M > 0$, and

$$Q(a, b) = \langle \phi(a, b), \theta^* \rangle$$

for all $(a, b) \in \mathcal{A} \times \mathcal{B}$.

To estimate the payoff matrix $Q$ from the observed data, our essential goal is to estimate $\theta^*$. Under Assumption 2.1, the linear system (1) can be rewritten as follows: for all $a \in \mathcal{A}$ and $b \in \mathcal{B}$,

$$\begin{cases} \langle (\phi(a, \cdot) - \phi(1, \cdot)) \nu^*, \theta \rangle = \log(\mu^*(a)/\mu^*(1))/\eta, \\ \langle (\phi(\cdot, b) - \phi(\cdot, 1))^\top \mu^*, \theta \rangle = -\log(\nu^*(b)/\nu^*(1))/\eta, \end{cases}$$

where $(\phi(a, \cdot) - \phi(1, \cdot)) \nu^*, (\phi(\cdot, b) - \phi(\cdot, 1))^\top \mu^* \in \mathbb{R}^d$. To simplify the notation, we define matrices

$$A(\nu) = ((\phi(a, \cdot) - \phi(1, \cdot)) \nu)_{a \in \mathcal{A}/\{1\}} \in \mathbb{R}^{(m-1) \times d},$$
$$B(\mu) = ((\phi(\cdot, b) - \phi(\cdot, 1))^\top \mu)_{b \in \mathcal{B}/\{1\}} \in \mathbb{R}^{(n-1) \times d},$$

and define vectors

$$c(\mu) = (\log(\mu(a)/\mu(1))/\eta)_{a \in \mathcal{A}/\{1\}} \in \mathbb{R}^{m-1},$$
$$d(\nu) = (-\log(\nu(b)/\nu(1))/\eta)_{b \in \mathcal{B}/\{1\}} \in \mathbb{R}^{n-1}$$

Then the linear constraints would be

$$\begin{bmatrix} A(\nu^*) \\ B(\mu^*) \end{bmatrix} \theta = \begin{bmatrix} c(\mu^*) \\ d(\nu^*) \end{bmatrix}. \tag{2}$$

Since the linear system has $m + n - 2$ constraints and the dimension of $\theta$ is $d$. Intuitively, if $d \le m + n - 2$ and the linear constraints are full rank, there is at most one solution of the above linear equations.

**Proposition 2.2** (Necessary and sufficient condition for strong identification). *Under Assumption 2.1, there is a unique $\theta \in \mathbb{R}^d$ such that $Q(a, b) = \langle \phi(a, b), \theta \rangle$ (i.e. $\theta = \theta^*$) for all $(a, b) \in \mathcal{A} \times \mathcal{B}$ if and only if the QRE satisfies the rank condition*

$$rank \left( \begin{bmatrix} A(\nu^*) \\ B(\mu^*) \end{bmatrix} \right) = d. \tag{3}$$

Let the rank condition (3) hold, so that the game is strongly identifiable. In an offline setting, we propose a two-step method to estimate $\theta^*$.

1. Estimate the QRE $(\mu^*, \nu^*)$ from the observed data and obtain $(\widehat{\mu}, \widehat{\nu})$.

2. Leverge (2) to estimate $\theta$. To be specific, we conduct the least-square estimation and obtain $\widehat{\theta}$:

$$\widehat{\theta} := \arg\min_{\theta \in \mathbb{R}^d} \left\| \begin{bmatrix} A(\widehat{\nu}) \\ B(\widehat{\mu}) \end{bmatrix} \theta - \begin{bmatrix} c(\widehat{\mu}) \\ d(\widehat{\nu}) \end{bmatrix} \right\|^2, \tag{4}$$

If the sample size is sufficiently large and $\text{TV}(\widehat{\mu}, \mu^*)$ and $\text{TV}(\widehat{\nu}, \nu^*)$ are close to zero, the coefficient matrix in (4) is of full column rank, and we can derive a closed form for $\widehat{\theta}$:

$$\widehat{\theta} = \left( \begin{bmatrix} A(\widehat{\nu}) \\ B(\widehat{\mu}) \end{bmatrix}^\top \begin{bmatrix} A(\widehat{\nu}) \\ B(\widehat{\mu}) \end{bmatrix} \right)^{-1} \begin{bmatrix} A(\widehat{\nu}) \\ B(\widehat{\mu}) \end{bmatrix}^\top \begin{bmatrix} c(\widehat{\mu}) \\ d(\widehat{\nu}) \end{bmatrix}. \tag{5}$$

Next, we derive the estimation error of the two-step method. Namely, given a finite sample bound for $\text{TV}(\widehat{\mu}, \mu^*)$ and $\text{TV}(\widehat{\nu}, \nu^*)$, we aim to derive $\|\widehat{\theta} - \theta^*\|$.

**Theorem 2.3** (Parameter estimation error). *Let $\epsilon_1$ and $\epsilon_2$ be two small numbers satisfying $\epsilon_1 < \min_{a \in [m]} \mu^*(a)$ and $\epsilon_2 < \min_{b \in [n]} \nu^*(b)$. Under Assumption 2.1 and the rank condition in (3), suppose $(\hat{\mu}, \hat{\nu})$ satisfies $TV(\hat{\mu}, \mu^*) \le \epsilon_1/2$ and $TV(\hat{\nu}, \nu^*) \le \epsilon_2/2$, then $\hat{\theta}$ constructed by (4) satisfies*

$$\|\hat{\theta} - \theta^*\|^2 \lesssim \epsilon_1^2 \cdot (1 + m \cdot (\epsilon_2^2 + 1)) + \epsilon_2^2 \cdot (1 + n \cdot (\epsilon_1^2 + 1)).$$

Now we present the finite sample result of the sample complexity. In the two-step method, given a dataset of agent actions following the true QRE, we first use a consistent estimator to approximate the true QRE and obtain $\widehat{\mu}, \widehat{\nu}$, then we use the estimated QRE to conduct the least square (5). Therefore, the sample complexity would be dependent on the convergence rate of the QRE estimator. A natural choice for QRE estimation is the frequency estimator.

**Theorem 2.4** (Finite sample error bound). *Given $N$ samples $\{(a^k, b^k)\}_{k \in [N]}$ following the true QRE $(\mu^*, \nu^*)$, we obtain $\widehat{\mu}, \widehat{\nu}$ by the frequency estimator. For any $\delta \in (0, 1)$, the estimation error bound of the payoff matrix holds with probability at least $1 - \delta$*

$$\|\widehat{Q} - Q\|_F^2 \lesssim \mathcal{O}\left( \frac{m^2 + n^2 + (m + n)\log(1/\delta)}{N} \right).$$

Theorem 2.4 provides a probabilistic guarantee for the accuracy of the reconstructed payoff matrix $\widehat{Q}$ in a finite-sample setting. The bound explicitly depends on the sample size $N$, the action space dimensions $m, n$, and the confidence parameter $\delta$. The estimation error decreases at a rate of $\mathcal{O}(1/N)$, which is consistent with the standard empirical result of the frequency estimator (van der Vaart, 1998). As the sample size $N$ increases, the errors of $\widehat{\mu}$ and $\widehat{\nu}$ decrease, leading to a more accurate reconstruction of the reward $Q^*$. On the other hand, the bound grows with the action space size in terms of $m^2 + n^2$, indicating that larger action spaces require more samples to achieve the same estimation accuracy.

### 2.3. Partial Identification

If the rank condition (3) does not hold, there are infinitely many $\theta \in \mathbb{R}^d$ that satisfy the QRE constraint (2). Under Assumption 2.1, the feasible set $\Theta \subset \mathbb{R}^d$ is

$$\Theta = \left\{ \theta : \begin{bmatrix} A(\nu^*) \\ B(\mu^*) \end{bmatrix} \theta = \begin{bmatrix} c(\mu^*) \\ d(\nu^*) \end{bmatrix}, \|\theta\| \le M \right\}.$$

Since the true parameter $\theta^*$ is partially identified, we construct a confidence set that contains the identified set with high probability. Given $N$ strategy pairs following the true QRE, we first estimate the QRE from the observed data by frequency estimators $\widehat{\mu}$ and $\widehat{\nu}$. Next, we select a threshold $\kappa_N > 0$ and construct the confidence set as follows:

$$\widehat{\Theta}_N = \left\{ \theta : \left\| \begin{bmatrix} A(\widehat{\nu}) \\ B(\widehat{\mu}) \end{bmatrix} \theta - \begin{bmatrix} c(\widehat{\mu}) \\ d(\widehat{\nu}) \end{bmatrix} \right\|^2 \le \kappa_N, \|\theta\| \le M \right\}. \tag{6}$$

To recover the feasible payoff functions, we simply compute $\widehat{Q}(a, b) = \phi(a, b)^\top \widehat{\theta}$ for all $\widehat{\theta} \in \widehat{\Theta}$ according to the linear assumption.

We demonstrate the effectiveness of our Algorithm by establishing its ability to construct accurate confidence sets. To be specific, we show that the confidence set $\widehat{\Theta}$ is close to the identified set $\Theta$ when the sample size $N$ is large. The key to approximating feasible set $\Theta$ is to identify a suitable threshold $\kappa_N$ that makes the confidence set $\widehat{\Theta}_N$ "similar" to $\Theta$. The following theorem formalizes this intuition.

**Theorem 2.5** (Convergence of confidence set). *Let Assumption 2.1 hold. For each $N \in \mathbb{N}$, suppose we observe $N$ samples $\{(a^k, b^k)\}_{k \in [N]}$ following the true QRE $(\mu^*, \nu^*)$, and calculate $(\widehat{\mu}, \widehat{\nu})$ by the frequency estimator. Set the confidence set $\widehat{\Theta}_N$ as in (6), where $\kappa_N = \mathcal{O}(N^{-1})$. Then with probability at least $1 - \delta$,*

$$d_H(\Theta, \widehat{\Theta}_N) \lesssim \frac{m + n + \sqrt{(m+n)\log(1/\delta)}}{\sqrt{N}}, \quad (7)$$

*where $d_H$ is the Hausdorff distance corresponding to the Euclidean distance in $\mathbb{R}^d$.*

Theorem 2.5 establishes the asymptotic consistency of our confidence set $\widehat{\Theta}_N$ in the finite-sample setting, showing that it converges to the true feasible set $\Theta$ as the number of observed samples increases. The finite-sample bound (7) demonstrates that the estimation error decreases at the rate of $\mathcal{O}(N^{-1/2})$, which matches the standard concentration rate for empirical frequency estimators. The dependence on $m$ and $n$ highlights that larger action spaces require more samples for the same level of confidence. This result confirms that our method provides both statistical consistency and a well-characterized finite-sample error bound, making it a robust approach for inverse game-theoretic inference.

### 2.4. Selection in Confidence Sets

As discussed in §2.3, the true parameter $\theta^*$ is partially identifiable when the rank condition (3) does not hold, and there are infinitely many parameters that lead to the same QRE. To avoid unnecessary large coefficients that might overfit or lead to instability, we define the optimal solution $\theta^*$ as the vector that satisfies the QRE constraints and has the minimum Euclidean norm, i.e.,

$$\theta^* = \operatorname*{argmin}_{\theta \in \mathbb{R}^d} \|\theta\|, \quad \text{subject to } \begin{bmatrix} A(\nu^*) \\ B(\mu^*) \end{bmatrix} \theta = \begin{bmatrix} c(\mu^*) \\ d(\nu^*) \end{bmatrix}.$$

When the system is not full column rank, the minimum-norm solution of least square is uniquely determined by the Moore–Penrose inverse (Ben-Israel & Greville, 2006):

$$\theta^* = \begin{bmatrix} A(\nu^*) \\ B(\mu^*) \end{bmatrix}^\dagger \begin{bmatrix} c(\mu^*) \\ d(\nu^*) \end{bmatrix}.$$

Therefore, to estimate the optimal parameter $\theta^*$, we propose the following plug-in estimator:

$$\widehat{\theta} = \begin{bmatrix} A(\widehat{\nu}) \\ B(\widehat{\mu}) \end{bmatrix}^\dagger \begin{bmatrix} c(\widehat{\mu}) \\ d(\widehat{\nu}) \end{bmatrix}.$$

Now we derive the estimation error bound $\|\widehat{\theta} - \theta^*\|$.

**Theorem 2.6** (Convergence of the optimal QRE solution). *Assume that the matrix*

$$X = \begin{bmatrix} A(\nu^*) \\ B(\mu^*) \end{bmatrix} \in \mathbb{R}^{(m+n-2) \times d}$$

*is of full row rank, and its smallest singular value is bounded from below, that is, $\sigma_{m+n-2}(X) \geq \sigma_b$ for some $\sigma_b > 0$. Given $N$ samples $\{(a^k, b^k)\}_{k \in [N]}$ following the true QRE $(\mu^*, \nu^*)$, we obtain $(\widehat{\mu}, \widehat{\nu})$ by the frequency estimator. For any $\delta \in (0, 1)$, when $N$ is sufficiently large, the following estimation error bound of the optimal QRE solution holds with probability at least $1 - \delta$:*

$$\|\widehat{\theta} - \theta^*\| \lesssim \frac{m + n + \sqrt{(m+n)\log(1/\delta)}}{\sqrt{N}}.$$

In practice, selecting the minimum-norm solution helps avoid overfitting and promotes stability (Hastie et al., 2009). The convergence rate $\mathcal{O}(N^{-1/2})$ matches standard results in statistical estimation, showing the reliability and efficiency of our method in practical settings.

## 3. Entropy-Regularized Zero-Sum Markov Games

In this section, we follow the same methodology in §2 and derive the inverse game theory for entropy-regularized two-player zero-sum Markov games.

### 3.1. Preliminary and Problem Formulation

We briefly review the setting of a two-player zero-sum Markov game (Littman, 1994), which is a framework that extends Markov decision processes (MDPs) to multi-agent settings, where two players with opposing objectives interact in a shared environment. A two-player zero-sum simultaneous-move episodic Markov game is defined by a sextuple $(\mathcal{S}, \mathcal{A}, \mathcal{B}, r, \mathbb{P}, H)$, where

- $\mathcal{S}$ is the state space, with $|\mathcal{S}| = S$,
- $\mathcal{A}$ and $\mathcal{B}$ are two finite sets of actions that players $i \in \{1, 2\}$ can take,
- $H \in \mathbb{N}$ is the number of time steps,
- $r = \{r_h\}_{h \in [H]}$ is a collection of reward functions, and
- $\mathbb{P} = \{\mathbb{P}_h\}_{h \in [H]}$ is a collection of transition kernels.

At each time step $h \in [H]$, the players 1 and 2 simultaneously take actions $a \in \mathcal{A}$ and $b \in \mathcal{B}$ respectively upon observing the state $s \in \mathcal{S}$, and then player 1 receives the reward $r_h(s, a, b)$, while player 2 receives $-r_h(s, a, b)$. Namely, the gain of one player equals the loss of the other. The system then transitions to a new state $s' \sim \mathbb{P}_h(\cdot | s, a, b)$ according to the transition kernel $\mathbb{P}_h$.

**Entropy-regularized two-player zero-sum Markov game.**
We study the two-player zero-sum Markov game with entropy regularization. We use $(\mu, \nu)$ to denote the policy of two players, where $\mu = \{\mu_h\}_{h=1}^H$ and $\nu = \{\nu_h\}_{h=1}^H$. At step $h$, the entropy-regularized V-function is

$$
V_h^{\mu,\nu}(s) = \mathbb{E}\Bigg[\sum_{t=h}^H \gamma^{t-h}\big[r_t(s_t, a_t, b_t) - \eta^{-1}\log\mu_t(a_t|s_t)
$$
$$
+ \eta^{-1}\log\nu_t(b_t|s_t)\big]\Big| s_h = s\Bigg],
$$

where $\gamma \in [0, 1]$ is the discount factor and $\eta > 0$ is the parameter of regularization. Meanwhile,, we define the entropy-regularized Q-function that

$$
Q_h^{\mu,\nu}(s, a, b) = r_h(s, a, b) + \gamma \mathbb{E}_{\mathbb{P}_h(\cdot|s,a,b)}\left[V_{h+1}^{\mu,\nu}(\cdot)\right]. \quad (8)
$$

For notation simplicity, we denote by $Q_h^{\mu,\nu}(s) \in \mathbb{R}^{m \times n}$ the collection of Q-functions at the state $s$, which is the matrix $[Q_h^{\mu,\nu}(s, a, b)]_{(a,b)\in\mathcal{A}\times\mathcal{B}}$. With this notation, we may write

$$
\begin{aligned}
V_h^{\mu,\nu}(s) &= \mu_h(s)^\top Q_h^{\mu,\nu}(s)\nu_h(s) \\
&+ \eta^{-1}\mathcal{H}(\mu_h(s)) - \eta^{-1}\mathcal{H}(\nu_h(s)).
\end{aligned} \quad (9)
$$

The equations (8) and (9) are also known as Bellman equations for Markov games. In a zero-sum game, one player seeks to maximize the value function while the other player wants to minimize it:

$$
V_1^*(s) = \max_\mu \min_\nu V_1^{\mu,\nu}(s) = \min_\nu \max_\mu V_1^{\mu,\nu}(s).
$$

**Definition 3.1** (Quantal response equilibrium). For each time step $h$, there is a unique pair of optimal policies $(\mu_h^*, \nu_h^*)$ of the entropy-regularized Markov game, i.e. the quantal response equilibrium (QRE), characterized by the following minimax problem:

$$
V_h^{\mu^*,\nu^*}(s) = \max_{\mu_h}\min_{\nu_h} V_h^{\mu,\nu}(s) = \min_{\nu_h}\max_{\mu_h} V_h^{\mu,\nu}(s).
$$

which is equivalent to

$$
\begin{aligned}
V_h^{\mu^*,\nu^*}(s) &= \max_{\mu_h}\min_{\nu_h} \mu_h(s)^\top Q_h^{\mu,\nu}(s)\nu_h(s) \\
&+ \eta^{-1}\mathcal{H}(\mu_h(s)) - \eta^{-1}\mathcal{H}(\nu_h(s)),
\end{aligned} \quad (10)
$$

where $\mu_h : \mathcal{S} \to \Delta(\mathcal{A})$ is the policy followed by player 1 and $\nu_h : \mathcal{S} \to \Delta(\mathcal{B})$ is the policy followed by player 2, and $\mathcal{H}(\pi) := -\sum_i \pi_i \log(\pi_i)$ denotes the Shannon entropy of a distribution $\pi$. Also, it is known that the unique solution of this minimax problem (QRE) satisfies the following fixed point equations:

$$
\begin{cases}
\mu_h^*(a|s) = \dfrac{e^{\eta\langle Q_h^*(s,a,\cdot),\nu_h^*(\cdot|s)\rangle_{\mathcal{B}}}}{\sum_{a\in\mathcal{A}} e^{\eta\langle Q_h^*(s,a,\cdot),\nu_h^*(\cdot|s)\rangle_{\mathcal{B}}}}, & \forall a \in \mathcal{A}, \\[4mm]
\nu_h^*(b|s) = \dfrac{e^{-\eta\langle Q_h^*(s,\cdot,b),\mu_h^*(\cdot|s)\rangle_{\mathcal{A}}}}{\sum_{b\in\mathcal{B}} e^{-\eta\langle Q_h^*(s,\cdot,b),\mu_h^*(\cdot|s)\rangle_{\mathcal{A}}}}, & \forall b \in \mathcal{B}.
\end{cases}
$$
$$(11)$$

**Goal.** We study the inverse game theory for this entropy-regularized two-player zero-sum Markov game, where both the rewards $(r_h)$ and the transition kernels $(\mathbb{P}_h)$ are unknown. To elaborate, we observe i.i.d. trajectories

$$
\{(s_1^t, a_1^t, b_1^t), \cdots, (s_H^t, a_H^t, b_H^t)\}_{t\in[T]}
$$

following the QRE $(\mu^*, \nu^*)$, and we aim to recover all the feasible reward functions $r$ defined as follows.

**Definition 3.2** (Identified reward sets). Given state and action space $\mathcal{S} \times \mathcal{A} \times \mathcal{B}$ and quantal response equilibrium $(\mu^*, \nu^*)$, a reward function $r : \mathcal{S} \times \mathcal{A} \times \mathcal{B} \to \mathbb{R}^H$ is identified if $\mu_h, \nu_h$ is the solution of the minimax problem (10) induced by the reward function $r_h$ for all $h \in [H]$.

### 3.2. Learning Reward Functions from Actions

In this section, we propose an algorithm to find all the feasible reward functions that lead to the QRE. We assume that both the reward function and transition kernel have a linear structure (Bradtke & Barto, 2004; Jin et al., 2020).

**Assumption 3.3** (Linear MDP). For the underlying MDP, we assume that for every reward function $r_h : \mathcal{S} \times \mathcal{A} \times \mathcal{B} \to [0, 1]$ and every transition kernel $\mathbb{P}_h : \mathcal{S} \times \mathcal{A} \times \mathcal{B} \to \Delta(\mathcal{S})$, there exist $\omega_h \in \mathbb{R}^d$ and $\pi_h(\cdot) : \mathcal{S} \to \mathbb{R}^d$ such that

$$
\begin{aligned}
r_h(s, a, b) &= \phi(s, a, b)^\top \omega_h, \\
\mathbb{P}_h(\cdot|s, a, b) &= \phi(s, a, b)^\top \pi_h(\cdot)
\end{aligned}
$$

for all $(s, a, b) \in \mathcal{S} \times \mathcal{A} \times \mathcal{B}$. In addition, the Q function is linear with respect to $\phi$. Namely, for any QRE $(\mu, \nu)$ and $h \in [H]$, there exists a vector $\theta_h \in \mathbb{R}^d$ such that

$$
Q_h(s, a, b) = \phi(s, a, b)^\top \theta_h.
$$

We assume $\|\phi(\cdot, \cdot, \cdot)\| \le 1$, $\|\theta_h\| \le R$, and $\|\pi_h(s)\| \le \sqrt{d}$ for all $h \in [H]$ and $s \in \mathcal{S}$.

*Remark* 3.4. In Assumption 3.3, since the reward functions $r_h$ are normalized to the unit interval $[0, 1]$ and the number of time steps $[H]$ is finite, every Q-function $Q_h$ must be bounded by some constant, and the constant $R \ge H(1 + \log m + \log n)$ exists. Since $(\omega_h)$ can be recovered by $(\theta_h)$, we prefer to make an assumption on $(\theta_h)$ instead of $(\omega_h)$ for the convenience of subsequent analysis.

We are going to find all the feasible $\omega_h$ for all $h \in [H]$ under Assumption 3.3. Analogous to matrix games, we first consider the identification problem of the Q-function. Namely, whether there is a unique $\theta_h$ corresponding to the QRE. Given the equilibrium constraint (11), we propose the following theorem for strong identification.

**Proposition 3.5** (Strong identification of Q-function). *Under Assumption 3.3, for each $h \in [H]$, the Q-function $Q_h(s, a, b) = \phi(s, a, b)^\top \theta_h$ is feasible for all $(s, a, b) \in$*

$\mathcal{S} \times \mathcal{A} \times \mathcal{B}$ if $\theta_h$ satisfies the following linear system:

$$\begin{bmatrix} A_h(s, \nu_h^*) \\ B_h(s, \mu_h^*) \end{bmatrix} \theta_h = \begin{bmatrix} c_h(s, \mu_h^*) \\ d_h(s, \nu_h^*) \end{bmatrix} \quad \text{for all } s \in \mathcal{S}, \quad (12)$$

where

$$A_h(s, \nu_h) = ((\phi(s, a, \cdot) - \phi(s, 1, \cdot)) \nu_h(\cdot|s))_{a \in \mathcal{A} \setminus \{1\}},$$
$$B_h(s, \mu_h) = ((\phi(s, \cdot, 1) - \phi(s, \cdot, b)) \mu_h(\cdot|s))_{b \in \mathcal{B} \setminus \{1\}}$$

and

$$c_h(s, \mu_h) = \left( \eta^{-1} \log \frac{\mu_h(a|s)}{\mu_h(1|s)} \right)_{a \in \mathcal{A} \setminus \{1\}} \in \mathbb{R}^{m-1},$$

$$d_h(s, \nu_h) = \left( -\eta^{-1} \log \frac{\nu_h(b|s)}{\nu_h(1|s)} \right)_{b \in \mathcal{B} \setminus \{1\}} \in \mathbb{R}^{n-1}.$$

Moreover, there exists a unique $\theta_h \in \mathbb{R}^d$ if and only if the QRE satisfies the rank condition

$$rank\left( \begin{bmatrix} A_h(\nu_h^*)^\top & B_h(\mu_h^*)^\top \end{bmatrix} \right) = d, \quad (13)$$

where

$$A_h(\nu_h) := \begin{bmatrix} A_h(1, \nu_h) \\ A_h(2, \nu_h) \\ \vdots \\ A_h(|\mathcal{S}|, \nu_h) \end{bmatrix}, \quad B_h(\mu_h) := \begin{bmatrix} B_h(1, \mu_h) \\ B_h(2, \mu_h) \\ \vdots \\ B_h(|\mathcal{S}|, \mu_h) \end{bmatrix}.$$

Following the Bellman equation (8), $r_h$ is a feasible reward function iff there exists a feasible Q function $Q_h$ and V function $V_{h+1}$ such that

$$r_h(s, a, b) = Q_h(s, a, b) - \gamma \mathbb{E}_{\mathbb{P}_h(\cdot|s,a,b)} [V_{h+1}(\cdot)]. \quad (14)$$

Next, we propose an algorithm to recover the feasible reward functions. For all $h \in [H]$, the algorithm performs the following four steps:

- Recover the feasible set by solving the least square problem associated with the linear system (12):

$$\widehat{\Theta}_h = \left\{ \|\theta\| \leq R : \left\| \begin{bmatrix} A_h(\widehat{\nu}_h) \\ B_h(\widehat{\mu}_h) \end{bmatrix} \theta - \begin{bmatrix} c_h(\widehat{\mu}_h) \\ d_h(\widehat{\nu}_h) \end{bmatrix} \right\|^2 \leq \kappa_h \right\}. \quad (15)$$

- Calculate the feasible Q and V functions ($Q_h$ and $V_h$) for all $\widehat{\theta}_h \in \widehat{\Theta}_h$.

- Estimate the transition kernel $\mathbb{P}_h$ from the observed data. Since the transition kernel has a linear structure, we employ ridge regression for estimation:

$$\Lambda_h = \sum_{t=1}^T \phi(s_h^t, a_h^t, b_h^t) \phi(s_h^t, a_h^t, b_h^t)^\top + \lambda \mathbf{I}_d,$$

$$\widehat{\mathbb{P}}_h \widehat{V}_{h+1}(s, a, b) = \phi(s, a, b)^\top \Lambda_h^{-1}$$

$$\times \sum_{t=1}^T \phi(s_h^t, a_h^t, b_h^t) \widehat{V}_{h+1}(s_{h+1}^t);$$

- Recover feasible set $\mathcal{R}_h$ by the Bellman equation (14).

### 3.3. Theoretical Guarantees

In this section, we present the theoretical results for our Algorithm. To begin with, we define the base metric to measure the distance between rewards.

**Definition 3.6** (Uniform metric for rewards). We define the metric $d$ between any pair of rewards $r, r'$ as

$$D(r, r') = \sup_{(h,s,a,b) \in [H] \times \mathcal{S} \times \mathcal{A} \times \mathcal{B}} |(r_h - r_h')(s, a, b)|.$$

We aim to recover the feasible reward set defined below.

**Definition 3.7** (Feasible reward set). We say a reward function $r = (r_1, r_2, \cdots, r_H)$ is feasible with respect to a quantal response equilibrium $\mu$ and $\nu$ if the Q function $Q = (Q_1, Q_2, \cdots, Q_H)$ satisfies the identifability condition (11) and the norm constraint $\|\theta_h^*\| \leq R$. We denote $\mathcal{R}$ as the feasible reward set corresponding to the quantal response equilibrium $\mu$ and $\nu$, namely,

$$\mathcal{R} := \left\{ r = (r_1, r_2, \cdots, r_H) : r \text{ is identified and} \right.$$

$$\left. \left\| \omega_h + \gamma \sum_{s \in \mathcal{S}} \pi_h(s) V_{h+1}(s) \right\| \leq R \text{ for all } h \in [H] \right\}.$$

Also, we denote $\mathcal{Q}$ as the feasible Q function set:

$$\mathcal{Q} = \{(Q_h)_{h=1}^H : Q \text{ is identified and } \|\theta_h\| \leq R, \forall h \in [H]\}.$$

Our formulation provides a principled way to handle partial identifiability in Markov games. Instead of forcing a single estimated reward function, we construct a structured set of feasible rewards, which offers a more robust approach to analyzing decision-making in complex multi-step strategic settings. Intuitively, the norm constraint $\|\theta_h\| \leq R$ plays a key role in ensuring that the estimated reward functions remain well-conditioned, and do not include arbitrarily large coefficients. Additionally, by linking the feasible reward set to the recursive Bellman equations (8)-(9), our definition ensures that every element of $\widehat{\mathcal{R}}$ maintains temporal consistency. In other words, the inferred rewards lead to equilibrium strategies that are valid over multiple decision-making steps.

For the sake of clarity, we fix the initial state distribution in the Markov game $\rho_1 \in \Delta(\mathcal{S})$, and define the marginal state visitation distributions associated with policies $\mu, \nu$ at each time step $h \in [H]$ as $d_h^{\mu,\nu}(s) = \mathbb{P}(s_h = s|\rho_1, \mu, \nu)$. Also, write the state-action visitation distributions as $d_h^{\mu,\nu}(s, a, b) = \mathbb{P}(s_h = s, a_h = a, b_h = b|\rho_1, \mu, \nu)$.

To control the uniform metric in Definition 3.6, we require an estimator of the QRE that performs uniformly well across

all states $s \in \mathcal{S}$. When using frequency estimators to approximate the policies $\mu_h^*(\cdot|s)$ and $\nu_h^*(\cdot|s)$, the estimation at each state is conducted independently. As a result, it is essential that the dataset sufficiently covers all states in $\mathcal{S}$ to obtain reliable estimates. To ensure this, we impose the following assumption, which guarantees that every state is visited with a minimum frequency throughout the horizon.

**Assumption 3.8** ($C$-well-posedness)**.** There exists a constant $C > 0$ such that

$$d_h^{\mu^*, \nu^*}(s) \geq C$$

for all $s \in \mathcal{S}$ and $h \in [H]$.

Now we are ready to present the theoretical results for the proposed algorithm.

**Theorem 3.9** (Sample complexity of constructing feasible reward set)**.** *Under Assumptions 3.3 and 3.8, let $\rho_h = d_h^*$ be the stationary distribution associated with optimal policies $\mu^*$ and $\nu^*$, where $h \in [H]$. We assume that the following $d \times d$ matrix*

$$\Psi_h = \mathbb{E}_{\rho_h} \left[ \phi(s_h, a_h, b_h) \phi(s_h, a_h, b_h)^\top \right]$$

*is nonsingular for all $h \in [H]$. Let $\mathcal{R}$ be the feasible reward set given in Definition 3.7. Given a dataset $\mathcal{D} = \{\mathcal{D}_h\}_{h \in [H]} = \{\{(s_h^t, a_h^t, b_h^t)\}_{t \in [T]}\}_{h \in [H]}$, we set $\lambda = \mathcal{O}(1)$, $\kappa_h = \mathcal{O}(T^{-1})$, and let $\widehat{\mathcal{R}}$ be the output of our Algorithm. Let $\xi = \min_{h \in [H], s \in \mathcal{S}, a \in \mathcal{A}, b \in \mathcal{B}} \{\mu_h^*(a|s), \nu_h^*(b|s)\}$. For any $\delta \in (0,1)$, let $T > 0$ be sufficiently large, so*

$$T \geq \max\left\{ \frac{1}{C^2} \log \frac{2HS}{\delta}, \frac{16(m \vee n)}{C\xi^2} \log \frac{4HS}{\delta}, \right.$$
$$\left. 512 \|\Psi_h^{-1}\|_{\mathrm{op}}^2 \log \frac{2Hd}{\delta}, 4\lambda \|\Psi_h^{-1}\|_{\mathrm{op}} \right\}.$$

*Then the following inequality holds with probability at least $1 - 3\delta$:*

$$D(\mathcal{R}, \widehat{\mathcal{R}}) \lesssim \frac{1}{\sqrt{T}} \left( \sqrt{S(m+n) \log \frac{HS}{\delta}} \log T \right.$$
$$\left. + S(m+n) \sqrt{\log \frac{HS}{\delta}} + \left( \sqrt{Sd} + \sqrt{d \log T} \right) \log(mn) \right),$$

*where $D$ is the Hausdorff distance corresponding to the uniform metric in Definition 3.6.*

Theorem 3.9 provides a strong guarantee on the accuracy of our reward recovery algorithm in Markov games. Our bound shows that the distance $D(\mathcal{R}, \widehat{\mathcal{R}})$ diminishes at the rate of $\mathcal{O}(T^{-1/2})$, which matches the optimal statistical rate for empirical risk minimization problems. This demonstrates that with sufficient data, the estimated reward functions remain close to the true feasible set, making our method

statistically reliable and sample-efficient. The explicit dependence on problem parameters offers insights into how exploration, feature representations, and action space size affect the difficulty of inverse reward learning in Markov games.

We also note that the condition that $\Psi_h$ is nonsingular ensures that the feature representation provides sufficient information for parameter recovery (Tu & Recht, 2017; Min et al., 2022). The norm $\|\Psi_h^{-1}\|_{\mathrm{op}}$ appears in the sample complexity bound, indicating that ill-conditioned feature matrices lead to larger estimation errors and require more samples to achieve the same level of accuracy.

In addition, instead of relying solely on frequency estimators for QRE estimation, we can extend our framework to integrate Maximum Likelihood Estimation (MLE) into our method and establish a convergence result with the same $T^{-1/2}$ rate.

# 4. Numerical Experiments

In this section, we implement our reward-learning algorithm and conduct numerical experiments in both entropy-regularized zero-sum matrix games and Markov games. All experiments are conducted in Google Colab. In this section we consider only two-player entropy-regularized entropy-regularized zero-sum Markov games.

**Setup.** We define the kernel function $\phi : \mathcal{A} \times \mathcal{B} \to \mathbb{R}^d$ with dimension $d = 2$, and set the true parameter $\omega_h$ that specifying reward functions to be

$$\omega_h^* = (0.8, -0.6)^\top$$

for all steps $h \in [H]$. We set the sizes of action spaces to be $m = 5$ and $n = 5$, the size of state space $S = 4$, and the horizon $H = 6$. The entropy regularization term is $\eta = 0.5$.

We implement the algorithm proposed in §3.2. In each experiment, our algorithm outputs a parameter $\widehat{\theta}_h$ in the confidence set $\widehat{\Theta}_h$. We set the bound of feasible parameters $\theta_h$ to be $R = 10$, and set the threshold $\kappa_h = 10^3/N$, where $N$ is the sample size. The regularization term in ridge regression is $\lambda = 0.01$.

**Metrics.** We evaluate the performance of our algorithm using two metrics: (1) the error in the estimated reward function $(\widehat{r}_h)$, which measures how accurately the reconstructed payoff function matches the true reward function; and (2) the error in the estimated QRE, which quantifies the discrepancy between the QRE $(\widehat{\mu}, \widehat{\nu})$ derived from the estimated payoff function and the true QRE $(\mu^*, \nu^*)$. We are particularly interested in the error in the estimated QRE, which validates whether the reconstructed reward functions interpret the observed strategy.

**Results.** As shown in Figures 1, 2 and Table 1, the overall error of our algorithm's output decreases as the sample size $N$ increases from $10^4$ to $10^5$, demonstrating the improved accuracy of our approach with more data. While the estimation error of reward functions $(\widehat{r}_h)_{h=1}^6$ can be relatively large, the corresponding QRE $(\widehat{\mu}_h, \widehat{\nu}_h)$ remains well-aligned with the true QRE $(\mu_h^*, \nu_h^*)$. Although some fluctuations are observed across time steps, the error remains small, especially for larger sample sizes. These results confirm that our method for reward estimation in Markov games is both statistically consistent and sample-efficient.

## 5. Conclusion

To conclude, we explore the challenge of recovering reward functions that explain agents' behavior in competitive games, with a focus on the entropy-regularized zero-sum setting. We propose a framework of inverse game theory concerning the underlying reward mechanisms driving observed behaviors, which applies to both the static setting (§2) and the dynamic setting (§3).

Under a linear assumption, we develop a novel approach for the identifiability of the parameter specifying the current-time payoff. To move forward, we develop an offline algorithm unifying QRE estimation, confidence set construction, transition kernel estimation, and reward recovery, and establish its convergence properties under regular conditions. Additionally, we adapt this algorithm to incorporate a MLE approach and provide theoretical guarantees for the adapted version. Our algorithms are reliable and effective in both static and dynamic settings, even in the presence of high-dimensional parameter spaces or rank deficiencies.

Future directions include exploring more complicated game settings, such as partially observable games and non-linear payoff functions, and extending the framework to online learning setting. Meanwhile, this research contributes to the broader effort to make competitive systems more interpretable, offering valuable insights at the intersection of game theory and reinforcement learning.

## Impact Statement

This work advances the field of inverse reinforcement learning and game theory by introducing a unified framework for reward function identification and estimation in competitive multi-agent settings. Our findings contribute to a deeper understanding of decision-making in strategic environments, with potential applications in economics, automated negotiation, and multi-agent AI systems.

While our research provides theoretical and methodological advancements, we acknowledge potential ethical considerations. The ability to infer reward functions from observed

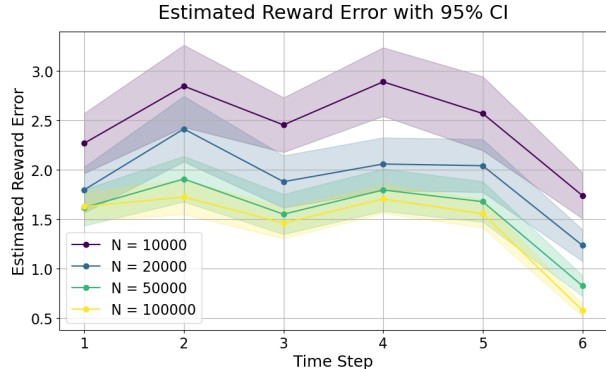

*Figure 1.* The reconstruction error of the reward functions $(\widehat{r}_h)_{h=1}^6$. The X-axis represents the time step $h$ from 1 to 6, while the Y-axis represents the error $\|\widehat{r}_h - r_h^*\|_{\mathrm{F}}$ of the reward function $\widehat{r}$.

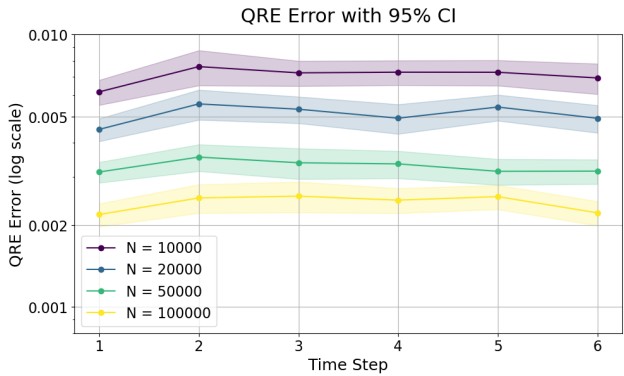

*Figure 2.* The discrepancy between the QRE $(\widehat{\mu}, \widehat{\nu})$ corresponding to the estimated reward functions $(\widehat{r}_h)_{h=1}^6$ and the true QRE $(\mu^*, \nu^*)$. The X-axis represents the time step $h$ from 1 to 6, while the Y-axis represents the errors $\mathrm{TV}(\widehat{\mu}_h, \mu_h^*) + \mathrm{TV}(\widehat{\nu}_h, \nu_h^*)$

| Sample Size | Reward Error | |
|---|---|---|
| | Mean | 95% CI |
| 10,000 | 2.4611 | $\pm 0.1596$ |
| 20,000 | 1.9031 | $\pm 0.1048$ |
| 50,000 | 1.5609 | $\pm 0.0663$ |
| 100,000 | 1.4398 | $\pm 0.0499$ |

| Sample Size | QRE Error | |
|---|---|---|
| | Mean | 95% CI |
| 10,000 | $7.08 \times 10^{-3}$ | $\pm 4.61 \times 10^{-4}$ |
| 20,000 | $5.11 \times 10^{-3}$ | $\pm 3.11 \times 10^{-4}$ |
| 50,000 | $3.28 \times 10^{-3}$ | $\pm 1.70 \times 10^{-4}$ |
| 100,000 | $2.41 \times 10^{-3}$ | $\pm 1.41 \times 10^{-4}$ |

*Table 1.* Mean error and 95% confidence intervals for reward and QRE estimation over 100 repetitions in the Markov game setting, across all time steps.

behavior could be used both positively—to enhance transparency in AI decision-making and improve algorithmic fairness—and negatively, if applied to manipulate or exploit agents in competitive settings. Ensuring the responsible application of this work will require careful consideration of ethical safeguards and alignment with societal values.

Overall, this paper aims to advance Machine Learning and Game Theory research, and we do not foresee immediate societal risks. However, we encourage further discussion on the ethical implications of inverse game theory in real-world applications.

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
