# OpenReview forum: "Decoding Rewards in Competitive Games: Inverse Game Theory with Entropy Regularization"
_ICML.cc/2025/Conference — ICML 2025 poster_

### Official Review · Reviewer_TbpT · 2025-03-09

**Overall Recommendation:** 4

**Summary:**

This paper studies inverse game theory for two-player zero-sum Markov games under entropy regularization (quantal response equilibrium): instead of best responding, each player plays a mixed strategy with softmax probability $\frac{e^{\eta u(a)}}{\sum_{a\in\mathcal A} e^{\eta u(a)}}$.  Given observed actions sampled from the quantal response equilibrium, this paper aims to estimate the utility function of the two players.  The paper derives the sample complexity of this problem, using a Maximum Likelihood Estimation approach.

**Claims And Evidence:**

All the theoretical and empirical results are solid as far as I can tell.

**Essential References Not Discussed:**

No in my knowledge.

**Experimental Designs Or Analyses:**

I don't see any issues with the experimental results.

**Methods And Evaluation Criteria:**

Using Maximum Likelihood Estimation to recover the utility functions in inverse reinforcement learning / inverse game theory under entropy regularization is a standard technique in the literature.  It makes sense for the problem at hand.

**Other Comments Or Suggestions:**

(1) The $\|\theta\| \le M$ in Assumption 2.1 should be $\|\theta^*\|\le M$.

(2) I would suggest not hiding essential parameters using the $\lesssim$ notation in the theorem statements (see "Theoretical Claims").

**Other Strengths And Weaknesses:**

### Strength:

(S1) Considering two-player games is a strength of this paper.  As far as I know, previous works on inverse reinforcement learning under entropy regularization mostly focus on single-agent problems.



### Weakness:

(W1) However, the two-player games considered by this paper are restricted to zero-sum games only.  Are there fundamental reasons that the approach in this work can only be applied to zero-sum games, or it can actually be applied to general-sum games?

(W2) Another big weakness is that the entropy-regularization parameter $\eta$ is assumed to be known by the learner.  In practice, different agents may use different regularization parameters and the parameters can be unknown.  When $\eta$ is known, as shown by the authors, the problem of estimating the utility function becomes a simple linear regression problem, which can be solved by standard MLE technique.  This technique doesn't seem to be applicable to unknown $\eta$.

**Questions For Authors:**

(Q1) What's the main difference between this work and previous works in IRL and inverse game theory with entropy regularization?  See "Relation to Broader Scientific Literature".

(Q2) Are there fundamental reasons that the approach in this work can only be applied to zero-sum games, or it can actually be applied to general-sum games?  (See Weakness 1)

(Q3) What if $\eta$ is unknown?  (See Weakness 2.)

(Q4) How do you derive the first equation in the proof of Theorem 2.3 (Line 665 - 670)?  In particular, how do you get $(A(\hat \mu)^\top A(\hat \mu) + B(\hat \mu)^\top B(\hat \mu))^{-1}$ from equation (5) ?

**Relation To Broader Scientific Literature:**

The key contribution of the paper to the broader literature is unclear.  The idea of inverse reinforcement learning under entropy regularization is well known and dates back to [Ziebart et al 2008](https://cdn.aaai.org/AAAI/2008/AAAI08-227.pdf).  More recent works about inverse game theory under entropy regularization include, for example, [Wu et al, 2024](https://arxiv.org/abs/2210.01380) and [Chen et al, 2023](https://arxiv.org/abs/2307.14085). The idea of using MLE to estimate the unknown reward functions for such problems is also standard.  The authors mentioned several related works in Section 1.2, but didn't mention the main difference between previous works and their work.  This makes it difficult to judge the contribution of this work.

**Theoretical Claims:**

I checked the proofs for the theorems in Section 2, which I believe are correct.



However, I have some concerns about the presentation of the theoretical results:  in particular, the asymptotic notation $\lesssim$ hides some important quantities that are not necessarily constants:

(1) The $\lesssim$ notatioin in Theorem 2.3 hide quantities $\eta, \epsilon_1, \epsilon_2$.  In particular, the $\frac{1}{\eta^2 (\min_{i\in[m]} \mu_i - \epsilon_1)^2}$ terms in equation (25) (Page 15) is hidden.  These quantities are important to the sample complexity when they are close to $0$.  They are not necessarily constants and should not be hidden by the $\lesssim$ notation in my opinion.

(2) Similarly, a quantity of $\sum_{a, b \in \mathcal A \times \mathcal B} \| \phi(a, b) \|^2$ is hidden by the $\lesssim$ notation in Theorem 2.4.  This quantity depends on the sizes of $\mathcal A$ and  $\mathcal B$, $m$ and $n$, which are not constants, and depends on $d$ which is not a constant, either.

 (3) Similarly, the $\lesssim$ notation in Theorem 2.5 hides $|| \Phi_1 ||$ and  $|| \Phi_2 ||$, which seem to depend on $m, n, d$, which are not constants.

Hiding too many such important quantities hurts the clarity and soundness of the theoretical results in my opinion.

---

> ### Author Rebuttal · Authors · 2025-03-31
>
> Dear Reviewer,
>
> Thank you for your valuable feedback and for carefully checking the proofs of our theorems.
> ## Concerns about the Asymptotic Notation
> We appreciate your observation regarding the use of asymptotic notation in Theorems 2.3, 2.4, and 2.5. To streamline presentation, we omitted terms that implicitly depend on the problem size—specifically, action space sizes $m,n$, feature dimension $d$, and the regularization parameter $\eta$. We agree that these quantities affect the sample complexity and convergence rates and are not truly constant. In the revised version, we will explicitly include these dependencies in the theorem statements and discuss their implications.
>
> ## Questions
> ### (Q1) Contribution to the Broader Litterature
> ```
> The idea of inverse reinforcement learning under entropy regularization is well known and dates back to Ziebart et al.
> ```
> Ziebart et al., 2008 focuses on imitation learning in **single-agent Markov Decision Processes**. In comparison, our work addresses IRL in two-player zero-sum Markov games, where the objective is to recover the reward functions given observed equilibrium strategies (QRE). Additionally, we deal with both strong and partial identifiability, which introduces new theoretical challenges not covered in Ziebart et al.
> ```
> More recent works about inverse game theory under entropy regularization include, for example, Wu et al and Chen et al.
> ```
> These studies focus on inverse game theory for **Stackelberg games** and **quantal Stackelberg equilibrium** (QSE). These methods primarily deal with leader-follower interactions, where one player commits to a strategy first, and the follower responds optimally. The difference between QRE and QSE is crucial:
> - QRE models simultaneous decision-making under entropy regularization, where both players make noisy best responses.
> - QSE models sequential decision-making, where the leader’s strategy is announced, and the follower best responds.
>
> Our theoretical framework leverages QRE constraint, which fundamentally differs from the QSE setting explored by Wu et al. and Chen et al.
> ```
> The idea of using MLE to estimate the unknown reward functions for such problems is also standard.
> ```
> We are sorry about causing the confusion. To clarify, we do not directly apply MLE for reward estimation. Instead, the key to our identification method is using the linear assumption to transform the QRE constraints into linear systems like (2), (11). We then employ least square to construct confidence sets. MLE is only used to estimate QRE from data (e.g., Appendix E), not to estimate rewards directly.
>
> To summarize, our contribution includes:
> - We introduce a **complete framework** that addresses reward identification and estimation in two-player zero-sum games and Markov games. To the best of our knowledge, no existing work addresses these specific problems.
> - We establish necessary and sufficient conditions for strong and partial identification under linear parameterization.
> - We provide theoretical guarantees for the sample complexity and convergence of the constructed confidence sets.
> - Through experiments, we verify the behavioral consistency between the recovered reward functions and the observed QRE.
>
>
> ### (Q2) Zero-Sum Game Restriction
> Our framework is currently specific to zero-sum games because of their **minimax structure**, which enables QRE to yield tractable linear constraints on the reward parameter. General-sum games lack this structure and may admit **multiple equilibria**, complicating both identification and estimation.
>
> Though extending our methods to general-sum settings is nontrivial, we view this as an exciting direction for future work and appreciate your suggestion to explore this aspect.
>
> ### (Q3) Assumption of Known Regularization Parameter $\eta$
> We acknowledge that our method assumes a known entropy regularization parameter $\eta$, which reflects our modeling of agents . In practice, agents may use different or unknown $\eta$'s. We address this in two parts:
> - If players use different $\eta_1$ and $\eta_2$, our linear formulation and results still hold by adjusting the vectors $c(\mu)$ and $d(\nu)$ accordingly.
> - If $\eta$ is unknown, one could treat it as a hyperparameter and select it via cross-validation. In estimation, the choice of $\eta$ is not a problem since we can scale the reward functions, so the identifiability and consistency of strategies remain unaffected.
>
> We will discuss this in the revised version as an important direction for future research.
>
> ### (Q4) Derivation in the Proof of Theorem 2.3
> Thank you for pointing out this issue. We apologize for the confusion caused by the notation. There was a minor typo in the proof where the inequality sign $\leq$ should have been $\lesssim$, which absorbs a constant factor of $2$. The estimate is a consequence of Cauchy-Schwartz inequality. We appreciate your careful attention to detail, and we will correct this in the final version of the paper.

---

> > ### Comment · Reviewer_TbpT · 2025-04-06
> >
> > Thank the authors for the response!  All of my concerns are resolved.  I would really appreciate it if the authors could clarify the asymptotic notations, highlight the contribution to the broader literature, acknowledge the limitations of zero-sum games and known $\eta$ assumptions, and improve the experimental results (as suggested by reviewers h55M and PH73) in the revised version.  I believe these improvements are easily doable, so I raised my score to 4.

---

> > > ### Author Response · Authors · 2025-04-06
> > >
> > > Dear Reviewer,
> > >
> > > Thank you very much for raising the score. Your thoughtful comments and insightful feedback are helpful in improving the quality of our work. We appreciate your suggestions regarding clarifying the asymptotic notations, highlighting our contribution to the broader literature, acknowledging the limitations of zero-sum games and known assumptions, and improving the experimental results as noted by reviewers h55M and PH73. We will incorporate these improvements in the final version.
> > >
> > > Thank you again for your kind support.

---

### Official Review · Reviewer_PH73 · 2025-03-13

**Overall Recommendation:** 3

**Summary:**

Under the assumption that agents are playing a QRE (a relaxation of Nash equilibrium that includes an entropy regularization term), this paper presents a method for learning the rewards of each agent based on a dataset of interactions in a finite-horizon zero-sum Markov game (or the special case of a single normal form game).  In the general case, the transition kernel must be learned in addition to the agent rewards.

The proposed method exploits a linearity assumption: that every action profile can be mapped to a finite-dimensional feature vector whose inner product with a weight vector yields the payoff of the maximizer agent. Based on this assumption and an empirical estimate of the agents' strategies from the observed interaction data, it produces an estimate of the weight vector $\theta$.

## Update after rebuttal
Thanks for your response!  In light of the rebuttal my opinion of the paper remains positive.

**Claims And Evidence:**

The theoretical claims appear well supported.  I do not find the empirical evaluation convincing.

**Essential References Not Discussed:**

[Chui et al. AAMAS 2023] perform empirical estimation of payoffs under a QRE assumption in the normal form setting.  I am not aware of any work which performs similar estimation in the Markov game setting.

**Experimental Designs Or Analyses:**

See "Methods"

**Methods And Evaluation Criteria:**

The empirical evaluations are on a single seemingly-arbitrary scenario, do not include checks of statistical significance, and do not appear to check the theoretical claims.  Answers to my "questions to authors" will provide insight into how serious these issues are.

**Other Comments Or Suggestions:**

* p.2 "with inner produce": inner product
* p.3: $d \le m + n - 2$ will not be true in general (in general there will be up to $mn$ unique utility pairs).  This is actually a pretty restrictive assumption.  The traveller's dilemma is a very structured game, and yet it has $3m$ unique utility profiles.  (It's not zero-sum though)
* p.4 "Let Assumption 2.1 and the rank condition (4) hold.": Do you mean the rank condition (3)?
* p.7 def.3.6 "between any pair of rewards $r,r' \in \mathcal{R}$": later on you define $\mathcal{R}$ as the set of feasible rewards corresponding to the QRE $\mu,\nu$; this is probably not what you mean here, because in Theorem 3.9 you want to bound distance away from $\mathcal{R}$, which doesn't make sense if the metric is only defined on $\mathcal{R}$.

**Other Strengths And Weaknesses:**

See "Questions"

**Questions For Authors:**

1. Theorem 3.9 "We assume that the following $d \times d$ matrix is nonsingular": What are the substantive implications of this assumption?  One is presumably that no feature is a linear transformation of another feature, is that sufficient, or does more need to be true?  What happens if this assumption fails to hold, does your whole method fail?  Can a user tell in advance that it fails to hold?


2. How did you choose the parameters of your test environment ($m=n=5$, $H=6$, etc.).

3. How many replications do the figures represent?  What are the confidence bounds on your plots?  Are the differences between $N=30000,N=100000,N=300000$ significant? (Especially in Figure 1)

4. Beyond the bare fact that more data means lower error, what should I take away from figures 1 and 2?  Are the quantitative differences in error in line with the theoretical bounds?

**Relation To Broader Scientific Literature:**

The related work is a solid survey of relevant literature.  I'm aware of work that aims to estimate payoffs in Stackelberg settings using the assumption of quantal response (some of which is cited).

**Theoretical Claims:**

No

---

> ### Author Rebuttal · Authors · 2025-03-31
>
> Dear Reviewer,
>
> Thank you for thoroughly evaluating our work and for your valuable comments. We hope our reply will address your concerns and questions.
>
> ### (Q1) Substantive Implications of the Non-singularity Assumption
> The non-singularity assumption on the feature covariance matrix $\Psi_h$ ensures that the feature vectors are linearly independent, meaning that no feature can be expressed as a linear combination of others, so we have sufficient information for parameter recovery. This is a standard assumption of coverage in many related literatures [1, Corollary 4.2; 2, Assumption 2.3]. One practical way for users to check this assumption is to examine the condition number of the matrix $\Psi_h$. If the condition number is extremely large, it indicates that the matrix is nearly singular. Additionally, principal component analysis (PCA) can help detect high collinearity among features.
>
> If the assumption does not hold, the problem becomes ill-posed, meaning that there exist multiple solutions for the estimated reward function. In such cases, our method may produce arbitrarily large or unstable estimates, as the problem becomes underdetermined. Nevertheless, in parctice,we can take proactive measures to avoid rank deficiency:
> - Reducing feature dimension to avoid over-parameterization.
> - Removing redundant features to ensure numerical stability and identifiability.
>
> ### (Q2) Parameter Choice
> We chose $m=n=5$, $H=6$ to strike a balance between model complexity and computational feasibility. The setup provides sufficient structure to validate both static and dynamic cases. We will add this explanation in the revised version.
>
> ### (Q3) Number of Replications and Confidence Bounds
> In the revised version, we repeat experiments 100 times and report 95% confidence intervals. Below we list the results on the QRE error:
>
> |Sample Size (T)|Mean Error and 95% Confidence Interval $(10^{-3})$|
> |-|-|
> |10,000|$7.55\pm 2.71$|
> |20,000|$5.36\pm 2.11$|
> |50,000|$3.40\pm 1.86$|
> |100,000|$2.27\pm 1.22$|
> We will include more result in our revised paper. This will ensure that the results are statistically robust and interpretable.
>
> In Figure 1, we compare reconstructed and assigned rewards. Since the true reward is not uniquely identifiable, different reward functions may induce the same QRE. Thus, even with large $N$, we may observe limited improvement in raw reward error. Our true goal is behavioral consistency, which we assess in Figure 2 via the QRE induced by the recovered reward. This metric better reflects algorithm performance and convergence.
>
> ### (Q4) Quantitative Differences and Theoretical Validation:
> Thank you for raising this important point. We acknowledge that the empirical results presented in the main text may not fully demonstrate consistency with the theoretical convergence rates.
>
> In fact, we did compare the empirical convergence rate with the theoretical result in the matrix game setting and verified their consistency (both $\mathcal{O}(N^{-1/2})$). This comparison and analysis are detailed in **Appendix E**. While we had limited Markov game replication before submission, we have since conducted 100-run experiments and observed consistent convergence rates. These additional results with confidence intervals will be added to the revision to strengthen empirical validation. We will update the revised version of the paper to include these new experimental results and confidence intervals, and explicitly discuss the consistency between empirical and theoretical results.
>
> Thank you once again for pointing out this important aspect. We believe that this additional empirical evidence will significantly strengthen the experimental validation of our theoretical findings.
>
> ### About Undiscussed Essential References
> Thank you for pointing out the recent work. While Chui et al. (2023) also explore preference estimation in games, their focus is fundamentally different from ours. They investigate non-strategic or **imperfectly rational players**, proposing models that better fit human behavior by relaxing equilibrium assumptions like Nash equilibrium and QRE. In contrast, our work assumes strategic agents playing QRE and studies the identification and estimation of reward functions under this equilibrium assumption. Moreover, Chui et al. do not address the identifiability or sample complexity of their estimation procedure, while a key contribution of our work lies in formally characterizing when the reward function is identifiable and how to estimate it reliably from finite data. Finally, their analysis is limited to static games, whereas our framework extends to finite-horizon Markov games.
>
> We will include a discussion of this work in the Related Work section.
>
> ### References
> [1] Tu, S. and Recht, B. (2017). Least-Squares Temporal Difference Learning for the Linear Quadratic Regulator.
>
> [2] Min, Y., Wang, T., Zhou, D. and Gu, Q. (2022). Variance-aware off-policy evaluation with linear function approximation.

---

### Official Review · Reviewer_h55M · 2025-03-14

**Overall Recommendation:** 4

**Summary:**

The paper analyzes the problem of identifying the utility in a zero-sum game from observations of a quantal response equilibrium policy. The authors provide an algorithm and theoretical guarantees on the recovered utilities. Moreover, they extend their analysis to zero-sum Markov games under a linear MDP assumption and provide experimental validation in a tabular setting.

**Claims And Evidence:**

The theoretical results seem sound, and they are presented clearly and convincingly.

**Essential References Not Discussed:**

While the authors briefly discuss single-agent IRL, they don't discuss the very much related results on identifiability in single-agent IRL. For instance, [1,2,3] discusses identifiability when learning from multiple experts with the same reward but individual transition laws. In particular, [1,2] frame reward identifiability problems as a subspace intersection problem and establish rank conditions similar to (12). Moreover, [3] establishes finite sample guarantees for both the feasible reward set and the suboptimality of $\pi_{\hat r}^*$ under the ground truth reward $r^*$. Given the similarities between your zero-sum formulation and learning from two distinct transition dynamics, providing a more detailed discussion that explicitly connects your findings to existing single-agent IRL results would enhance the theoretical clarity and positioning of your work.

1) Cao, Haoyang, Samuel Cohen, and Lukasz Szpruch. "Identifiability in inverse reinforcement learning." Advances in Neural Information Processing Systems 34 (2021)
2) Rolland, Paul, et al. "Identifiability and generalizability from multiple experts in inverse reinforcement learning." Advances in Neural Information Processing Systems 35 (2022)
3) Schlaginhaufen, Andreas, and Maryam Kamgarpour. "Towards the transferability of rewards recovered via regularized inverse reinforcement learning." Advances in Neural Information Processing Systems 35 (2024)

**Experimental Designs Or Analyses:**

The authors provide an experimental validation in the tabular case. They show that both rewards and the corresponding QRE are close to the expert's reward and QRE. I have the following suggestions:
1) Clarify whether the rank condition (12) is satisfied in the current experimental setup, as only then could we expect reward identifiability.
2) Discuss whether closeness of QREs is theoretically expected. Although your current analysis seems to lack a formal continuity argument, I'd expect that it would hold in the entropy-regularized setting.
3) Repeat experiments and add confidence bars to your plots.

**Methods And Evaluation Criteria:**

The paper proposes first estimating the expert policies, followed by estimating a confidence set for the payoff or Q function parameter via least-squares. Additionally, in the linear MDP setting, the transition kernel is estimated, and a confidence set for the reward is recovered via the Bellman equation. This methodology is sound and consistent with approaches from prior work in the single-agent setting.

**Other Comments Or Suggestions:**

Minor issues & typos:
- line 27, right column: ", as and multiple ..."
- line 192, right column: "approximating feasible set..."
- line 207, left column: "frequency estimator" not defined explicitly.
- line 330 & 343, left column: "estimating the transition kernel" is mentioned twice.

**Other Strengths And Weaknesses:**

The theoretical results are interesting, sound, and clearly presented. However, the paper would benefit from some more explanations of these theoretical results to help build intuition, e.g. when is the rank condition satisfied, what does $D(\mathcal{R},\hat{\mathcal{R}})<\varepsilon$ imply in practice?

**Questions For Authors:**

1. Can you give any theoretical insights on when the rank condition (12) is satisfied? Say in a tabular setting.
2. Can you say something about the QREs corresponding to your recovered reward?

**Relation To Broader Scientific Literature:**

The paper builds on ideas from both single-agent reinforcement learning and inverse game theory. In single-agent IRL, similar identifiability conditions have been established for learning from multiple experts with the same objective but different transition kernels (see references below).

**Theoretical Claims:**

While I didn't have the time to check all the proofs, the claims made seem reasonable and in line with what we would expect from the single-agent theory.

---

> ### Author Rebuttal · Authors · 2025-03-31
>
> Dear Reviewer,
>
> Thank you for your thoughtful and constructive feedback! We greatly appreciate your positive assessment of the theoretical soundness and methodological consistency of our work. Below, we address your specific concerns and suggestions.
>
> ### Relation to Single-Agent IRL Identification
> We appreciate the suggestion to connect our results with identifiability in single-agent IRL. In particular, works such as Cao et al. [1], Rolland et al. [2], and Schlaginhaufen & Kamgarpour [3] explore how observing behavior under varying transition dynamics or discount factors can lead to identifiability, typically through rank or subspace conditions. These insights align with our own setting, where we identify necessary and sufficient rank conditions on linear systems derived from QRE constraints in zero-sum Markov games.
>
> Unlike the single-agent case, we address multi-agent, strategic interactions where agents follow a quantal response equilibrium. Nevertheless, the underlying intuition is similar: entropy regularization and structured assumptions (e.g., linear parameterization) enable us to derive strong identifiability and sample-efficient estimation. We will incorporate this discussion in the Related Work section to better position our contributions.
>
> ### Experimental Design
> We have now conducted additional experiments in the Markov game setting with 100 replications per sample size and computed 95% confidence intervals. These results align closely with our theoretical rates. Below are some results on the QRE error $\mathrm{TV}(\widehat{\mu},\mu)+\mathrm{TV}(\widehat{\nu},\nu)$:
>
> |Sample Size (T)|Mean Error and 95% Confidence Interval $(10^{-3})$|
> |-|-|
> |10,000|$7.55\pm 2.71$|
> |20,000|$5.36\pm 2.11$|
> |50,000|$3.40\pm 1.86$|
> |100,000|$2.27\pm 1.22$|
> We will update our paper to include these results and clearly explain the statistical significance of the results.
>
> ### (Q1) Theoretical Insights on When the Rank Condition (12) Is Satisfied
> The rank condition (12) ensures the linear system (11) has a unique solution, which is the strongly identifiable case. Here the matrices $A_h(s,\nu_h^*)$ and $B_h(s,\mu_h^*)$ are derived from differences between feature vectors weighted by the corresponding equilibrium strategies, which means that distinct action pairs should lead to linearly independent differences. The rank condition also implicitly requires that the QRE policies $(\mu_h^*,\nu_h^*)$ do not degenerate and effectively captures the difference between different action pairs.
>
> In the tabular case, where $\phi$ is the canonical map and $d=Smn$, this condition does not hold in general, since the system (11) has only $S(m+n-2)$ equations. Nevertheless, our theory for the partially identifiable case ensures that we can still learn an acceptable estimate for the reward function even when the rank condition is not satisfied.
>
> In practice, we recommend using low-dimensional, non-redundant feature maps and applying tools like PCA to remove collinearity, which helps maintain full rank and improves sample efficiency.
>
> ### (Q2) QREs Corresponding to the Recovered Reward
> Theoretically, our algorithm aims to recover a confidence set of feasible reward functions rather than a single one, due to the partial identifiability inherent in inverse game learning. In practice, our focus is on behavioral consistency rather than exact reward matching.
>
> In the entropy-regularized setting, small variations in rewards result in proportionally small variations in QRE, according to the Lipschitzness of softmax. Consequently, when the error between the recovered reward and the true reward is small, the QRE induced by the recovered reward function is guaranteed to be close to the true QRE.
>
> Our experimental results consistently show that even when the recovered reward function differs from the assigned one (due to partial identifiability), the QRE derived from the recovered reward still closely matches the observed QRE. This indicates that **the estimated reward is close to another feasible reward**.
>
> ### Other Strengths And Weaknesses
> Thank you for your positive feedback on the theoretical results and for suggesting ways to build more intuition. Below, we address the specific point you raised.
>
> The distance measure quantifies the Hausdorff distance between the estimated feasible set $\widehat{\mathcal{R}}$ and the true feasible set $\mathcal{R}$. In practice, when it is sufficiently small, we know that
>
> - The estimated reward set $\widehat{\mathcal{R}}$ almost captures all feasible rewards $r\in\mathcal{R}$;
> - For any estimated reward $\widehat{r}$, there exists a feasible reward $r$ close to it, so the confidence set $\widehat{\mathcal{R}}$ is not too conservative.
>
> If the rank condition (12) holds, our confidence set $\widehat{\mathcal{R}}$ converges to a single point, which is the uniquely feasible reward $r$. Indeed, even if the rank condition does not hold, our algorithm still efficiently recovers all feasible reward functions.

---

### Official Review · Reviewer_Jg64 · 2025-03-24

**Overall Recommendation:** 3

**Summary:**

The submission considers inverse reinforcement learning in games. Specifically, the authors first study the conditions for the problem to identifiable, and the propose a methodology to estimate the reward function. Both theoretical analysis and empirical results are provided to justify the proposed method.

**Claims And Evidence:**

The claim that that the proposed condition is necessary is not well discussed (proposition 2.2). First of all, proposition 2.2 does not explicitly claim the condition to be necessary. This itself is contradictory the previous claims and the subtitle of the proposition. Second, to my understanding, the proposed condition is necessary only under some assumptions like the model setup. Specifically, there are similar works (IRL while not in the game setting) that does not take a linear condition like [1,2,3].

The discussions regarding why the condition is necessary and on the comparisons with the mentioned works are missing.

[1] Reward Identification in Inverse Reinforcement Learning
[2] Reward-Consistent Dynamics Models are Strongly Generalizable for Offline Reinforcement Learning
[3] Deep PQR: Solving Inverse Reinforcement Learning using Anchor Actions

**Essential References Not Discussed:**

Already mentioned above.

**Experimental Designs Or Analyses:**

I did not pay very much attention to this. The design makes sense to me but lacks competing methods.

**Methods And Evaluation Criteria:**

I am not farmiliar enough to the direction of IRL for games to suggest any competing methods in this line. But the submission does not consider any competing methods in the experiment section. As a result, the the advantage of the proposed method is not well justified.

**Other Comments Or Suggestions:**

NA

**Other Strengths And Weaknesses:**

NA

**Questions For Authors:**

Already mentioned above.

**Relation To Broader Scientific Literature:**

The submission contributes to the area of inverse reinforcement learning for games. For such problems, a key challenge is identifiablity. The submission uses a linear-like condition to solve this issue.

**Theoretical Claims:**

I did not check the proof of the results. I tried to find the proof for proposition 2.2 especially on why the proposed condition is necessary, but did not find the proof.

In general the proof makes sense to me: by posing the linear assumption, the reward identification is indeed an MLE problem, whose theoretical guarantees are well established even given not perfectly IID data.

---

> ### Author Rebuttal · Authors · 2025-03-31
>
> Dear Reviewer,
>
> Thank you for your valuable feedback and for recognizing the relevance and significance of our contributions. Below, we address your concerns regarding the necessity of the rank condition (Proposition 2.2), comparisons with related works, and competing methods in experiments.
>
> #### Necessity of the Rank Condition (Proposition 2.2)
> We apologize for our unclear statement in Proposition. The submitted version of the paper may not clarify the necessary conditions and their assumptions. In fact, the reward parameter is uniquely solvable **if and only if** the rank condition holds, and the logic is derived in previous contents.
>
> - $(\mu^*,\nu^*)$ is the quantal response equilibrium (QRE) corresponding to $Q$ if and only if the following QRE constraint is satisfied:
> $$
> \mu^*(a)= \frac{e^{\eta Q(a,\cdot)\nu^*}}{\sum_{a\in\mathcal{A}}e^{\eta Q(a,\cdot)\nu^*}}\ \text{for all}\ a\in\mathcal{A},\quad
> \nu^*(b)= \frac{e^{-\eta  Q(\cdot,b)^\top\mu^*}}{\sum_{b\in\mathcal{B}}e^{-\eta  Q(\cdot,b)^\top\mu^*}}\ \text{for all}\ b\in\mathcal{B}.
> $$
> - Under the linear assumption, the above non-linear system is equivalent to the linear system (2).
> - According to our model assumption, this linear system has at least one solution $\theta=\theta^*$. The solution is unique if and only if the rank condition (3) is satisfied.
>
> We will correct our statement in a later version. We hope this clarification solves your confusion.
>
> #### Comparisons with Prior Works
> We understand your concern regarding related works that do not impose a linear condition. The works [1,2,3] you listed explore nonlinear settings and adopt different **model assumptions**. In particular:
>
> [1] (Reward Identification in Inverse Reinforcement Learning) studies the problem of reward identifiability in IRL from a **graph** perspective, focusing on single-agent MDPs. It primarily addresses identifiability conditions **without discussing the problem of reward estimation in identifiable cases.**
>
> [2] (Reward-Consistent Dynamics Models for Offline Reinforcement Learning) focuses on **model-based offline RL**, where the goal is to learn a dynamics model that generalizes well to unseen transitions. It is applied to single-agent settings, and **the problem of reward identification is not addressed.**
>
> [3] (Deep PQR: Solving Inverse Reinforcement Learning using Anchor Actions) works on inverse reinforcement learning (IRL) with **deep energy-based policies**. The key innovation is the introduction of an anchor action, which is a known, zero-reward action (e.g., doing nothing), to facilitate reward identification. Compared with our linear assumption, this method is based on an Anchor-Action Assumption (See Assumption 1), which helps to uniquely determine the reward function.
>
> These works differ significantly from our approach and share limited similarities with our problem setting. In our work, the linearity assumption is crucial to transform the non-linear QRE constraint into a tractable linear problem. This enables us to derive rigorous theoretical guarantees on both identification and estimation of reward functions. Moreover, we provide a thorough theoretical analysis including convergence guarantees and confidence set construction, which is not covered in above works. We will include a more detailed discussion of these related works in the final version. Thank you for pointing out these references and helping us improve the clarity of our presentation.
>
> #### Lack of Competing Methods
> We would like to clarify that our paper introduces a **totally novel framework** for inverse game theory, covering both identification and estimation of reward functions in competitive game settings.
> To the best of our knowledge, **there are no existing methods specifically designed for inverse reinforcement learning (IRL) in two-player zero-sum Markov games that are directly comparable to our approach**. Most prior works in IRL focus on single-agent settings or non-competitive scenarios. Furthermore, the entropy-regularized QRE setting that we address is particularly unique, as it requires handling the complexity of equilibrium strategies and partial identifiability.
>
> While there are some works that address inverse RL with deep policies or anchor actions (e.g., [3]) or model-based RL for offline settings (e.g., [2]), these methods do not address the competitive, game-theoretic setting we focus on. Our approach leverages linear parametrization and QRE modeling, which makes it fundamentally different from existing IRL techniques that do not incorporate game-theoretic interactions or entropy regularization.
>
> We would greatly appreciate it if the reviewer could suggest any closely related methods that might have been overlooked or not covered in our paper. We are committed to enhancing our work by incorporating relevant comparisons, if available, and we value your input in guiding us toward potential improvements.

---

### Decision · Program_Chairs · 2025-05-01

**Decision:**

Accept (poster)

**Comment:**

This paper studies the problem of inverse reinforcement learning in games by observing demonstrations of the quantal response equilibrium in two-player zero-sum normal form games as well as Markov games. All the reviewers were enthusiastic about the technical contribution of this paper, and so I am recommending acceptance. Multiple reviewers mention that contextualization of the result with related work (e.g. in single-agent IRL) could be much improved. I recommend the authors pay close attention to the many helpful suggestions made by the reviewers to improve their related-work exposition for the camera-ready version, as well as better justify their assumptions.